# Blood Pressure Gradients in the Brain: Their Importance to Understanding Pathogenesis of Cerebral Small Vessel Disease

**DOI:** 10.3390/brainsci9020021

**Published:** 2019-01-23

**Authors:** J. David Spence

**Affiliations:** Stroke Prevention & Atherosclerosis Research Centre, Robarts Research Institute, Western University, 1400 Western Road, London, ON N6G 2V4, Canada; dspence@robarts.ca; Tel.: +1-519-931-5731; Fax: +1-519-931-5737

**Keywords:** Small vessel disease, lacunar infarction, hypertension, cerebral autoregulation, blood pressure gradients, vascular centrencephalon

## Abstract

The term “lacunar infarction” referred to small infarctions in the basal ganglia, internal capsule, thalamus, and brainstem, due to hypertensive small vessel disease. However, it has become common to refer to all small infarctions as lacunar. It is important to understand that true lacunes occur in a phylogenetically ancient part of the brain, the “vascular centrencephalon”, where short straight arteries with few branches transmit high blood pressure straight through to end-arterioles. The cortex is supplied by long arteries with many branches, so there is a very large blood pressure gradient in the brain. When blood pressure in the brachial artery is 117/75 mmHg, the pressure in the lenticulostriate artery would be 113/73, and the pressure in small parietal arterioles would be only 59/38 mmHg. Recent studies have reported that patients with a pulse pressure >60 mmHg and diastolic pressure <60 mmHg have a doubling of coronary risk and a 5.85-fold increase in stroke risk. This means that new low systolic targets being proposed will probably decrease the incidence of true lacunes, but increase small subcortical infarctions in the hemispheres. The pathogenesis of small vessel disease should be interpreted in the light of these blood pressure gradients.

## 1. Introduction

There is a widespread tendency to assume that small vessel disease in the brain (ischemia due to arteriolar disease, as opposed to atherosclerosis or emboli) is one single condition, but it is apparent that there are several different kinds of small vessel diseases. The purpose of this review is to focus on an aspect of small vessel disease that is seldom considered: Blood pressure gradients in the brain. Apart from vasculitis and rare conditions such as Cerebral Autosomal Dominant Arteriopathy with Subcortical Infarcts and Leukoencephalopathy (CADASIL), it appears likely that there are at least three different kinds of small vessel disease, found in different regions of the brain: Deep/hypertensive, lobar/hypotensive, and periventricular/venous.

It is clear that a stroke is a major contributor to dementia. This is particularly the case for the kinds of strokes associated with small vessel disease. The association goes in both directions: Strokes aggravate dementia, and amyloid (which is associated with Alzheimer disease) aggravates strokes.

In the Nun Study [1], a prospective study of dementia in residents of convents in the US, the participants had cognitive testing during life, and consented to autopsy at the time of their death. Many of the participants who had marked pathological changes of Alzheimer’s disease at autopsy had not yet experienced cognitive decline; however, the likelihood that the participants had been demented during life was increased 20-fold by even one or two small infarctions at the base of the brain. In a rat model, Whitehead et al. studied infarct size and cognitive dysfunction, comparing ischemia alone, with ischemia and injection of Beta amyloid. They found that infarct volumes were higher in the presence of amyloid compared with the ischemia alone. With ischemia alone, infarct volume was significantly smaller 28 days after surgery, compared with 7 days after surgery. However, when amyloid was added to ischemia, the infarct volume was significantly larger 28 days after surgery, compared with 7 days after surgery. Neuro-inflammation in the region of the infarct was also significantly increased. The Barnes circular platform test showed that time-dependent increases in memory and learning deficits were greater when beta-amyloid treatment was combined with ischemia. Markedly increased inflammation was found in the region of infarctions, the size of infarctions, and impaired cognitive function after injection of endothelin [2]. The mechanisms underlying the interactions of stroke, dementia, amyloid, and inflammation were reviewed in 2014 [3].

## 2. Misuse of the Term “Lacunar Infarction”

When C. Miller Fisher initially described the lacunar syndromes in relation to hypertensive small vessel disease [4,5,6], the term was intended to refer to infarctions at the base of the brain, where blood pressure is high. He referred to small infarctions due to hypertensive small vessel disease: Occlusion of small arterioles, due to hyaline degeneration and fibrinoid necrosis. Micro-atheroma has since been added to the pathogenesis of lacunar stroke.

That region of the brain was named by Hachinski the “Vascular Centrencephalon” [7]; the phylogenetically ancient part of the brain that is perfused by short straight arteries with few branches, transmitting pressure over a short distance directly from the large arteries to end-arterioles. When the small arterioles occlude, the result is a lacunar infarction; when they rupture, the consequence is intra-cerebral hemorrhage. This is illustrated in Figure 1. Hypertension causes hemorrhages in that distribution—in the brainstem, basal ganglia, internal capsule, and thalamus. Lobar hemorrhages are not due to hypertensive small vessel disease, but due to amyloid angiopathy [8].

Since the advent of magnetic resonance imaging (MRI,) it has become clear that small white matter intensities are highly predictive of cognitive decline. It has become increasingly common to refer to all small infarctions (usually <1.5 cm) as “lacunar”, but this mis-nomerization has led to confusion about the pathogenesis of small vessel disease. This is important because it may confuse efforts in the prevention of dementia via the prevention of stroke. It is clear that treating hypertension is one way to prevent/delay dementia; this is no doubt due to the prevention of hypertensive small vessel disease. Recent guidelines have recommended lowering the definition of hypertension to a systolic pressure <130 mmHg; however, with even lower blood pressure targets such as 120 mmHg being contemplated in the wake of the Systolic Pressure Intervention Trial (SPRINT), [9] there is reason to be concerned about a possible increase in the risk of strokes in the small sub-cortical vessels, over the vertex of the hemispheres.

## 3. Blood Pressure Gradients in the Brain

The human cortex is perfused by long arteries with many branches, resulting in a marked decrease in blood pressure between the vascular centrencephalon and the small arterioles perfusing subcortical regions over the convexity. Blood pressure drops as flow is divided among branches, so at the distal end of the long arteries with many branches, blood pressure is much lower than in the parent artery. Blanco et al. [11] calculated that, when blood pressure in the brachial artery is 192/113 mmHg, the pressure in the small arterioles of the posterior parietal artery bed would be only 117/68 mmHg. In the normotensive case, with blood pressure in the brachial artery of 117/75 mmHg, the pressure in the lenticulostriate artery would be 113/73, the pressure in the arterioles of the lenticulostriate bed (30–50 µM) would be 91/58, but in the same-sized small parietal arterioles would be only 59/38 mmHg. In the hypertensive case, when blood pressure in the brachial artery is 192/113 mmHg, the pressure in the lenticulostriate artery would be 183/110, in the small arterioles of the lenticulostriate bed, it would be 169/101 mmHg, and in the small arterioles of the posterior parietal artery bed, it would be only 117/68 mmHg. (Figure 2) This under-recognized phenomenon has great importance for understanding small vessel disease, because it means that the pathogenesis of true lacunar infarctions in the high-pressure regions of the brain must be different from that causing small infarctions in the sub-cortical regions over the convexity. 

## 4. Diastolic J Curve with Wide Pulse Pressure

There is a particular sub-group of patients who would be at risk from a systolic blood pressure target of 120 mmHg: Those with stiff arteries, causing a wide pulse pressure. Two important papers have shown a marked diastolic J curve in patients with a pulse pressure >60 mmHg and a diastolic pressure <60 mmHg. McEvoy et al. [12] reported from the Atherosclerosis Risk in Communities (ARIC) study a doubling of myocardial ischemia in such patients. Park and Ovbiagele [13] reported from an analysis of data from the Vitamin Intervention in Stroke Prevention (VISP) trial [14] a 5.85-fold increase in the risk of stroke. There are at least two explanations for this phenomenon. Firstly, virtually all myocardial perfusion occurs during diastole, and more than half of cerebral perfusion is during diastole (that is why the characteristic ultrasound signature of the internal carotid artery is a much higher diastolic velocity compared to systemic arteries) [15]. Secondly, patients with a wide pulse pressure have stiff arteries, and are more likely to have a large cuff artefact in measurement of their blood blood pressure by a cuff method. 

## 5. Cuff Artefact in Measurement of Blood Pressure

Spence reported in 1978 [16] that among patients age > 60 with diastolic blood pressures >100 mmHg, but no hypertensive end-organ disease, half had a cuff blood pressure in the brachial artery, that was >30 mmHg higher than the intra-arterial pressure measured in the distal brachial artery. This happens because the cuff must squeeze harder to collapse stiff arteries. Osler commented on this, and Messerli studied the Osler Maneuver (palpability of a pulseless radial artery) in relation to cuff artefact. In 1985 our group reported that patients with stiff arteries, identified by a high pulse-wave velocity, were more likely to have a large cuff artefact [17]. Thus patients with stiff arteries are at risk of diastolic pressures that are too low, with a systolic target of 120 mmHg. Not only will their cuff pressure be below 60 mmHg; it is possible, and even likely, that the true intra-arterial pressure may be substantially lower. 

The problem will be aggravated by beta-blockers and perhaps diltiazem, because bradycardia results in an increased stroke volume, which widens pulse pressure in patients with stiff arteries. 

The distribution of deep versus lobar white matter intensities (WMI), and their relationship to hypertension and pulse pressure, are illustrated in a paper from the group of Warlaw in Edinburgh [18]. Symptomatic deep WMI were related to hypertension, whereas, asymptomatic lobar WMI were more closely related to pulse pressure.

## 6. Impaired Autoregulation of Cerebral Blood Flow

A special form of this problem occurs with CADASIL, with “impaired vascular reactivity in sub-cortical vessels, in part due to arterial changes of thickened hyalinized walls, smooth muscle degeneration, and accumulation of granular osmiophilic material (GOM). This “earthen pipe” state of vessels impairs auto-regulation, with dependence on systemic blood pressure for perfusion of sub-cortical regions.” Chronic hypo-perfusion can accelerate cognitive decline in patients with CADASIL. [19] Acute hypotension can also precipitate infarction in CADASIL. Pettersen et al. described a case of serial severe infarctions in a patient with CADASIL, following episodes of acute hypotension after a motor vehicle collision. [20]

Patients with longstanding hypertension have a thickening of their cerebral arterioles, due to hypertrophy of the muscle layer, the result of constricting against higher pressure. They therefore, have a shift of their cerebral blood flow auto-regulation to higher pressures, so they are more likely to have strokes if blood pressure is suddenly dropped, for example with such therapies as “sub-lingual” nifedipine, or intra-muscular hydralazine [21]. (Figure 3). The other end of this spectrum is that patients with severe carotid stenosis have thinning of their cerebral arterioles. Those with severe bilateral carotid stenosis are protected from strokes due to hypertension [22], but patients with severe stenosis may be at risk of cerebral hyper-perfusion syndrome [23] (analogous to hypertensive encephalopathy) if blood pressure is high, following endarterectomy or stenting.

## 7. Venous Small Vessel Disease

A further complication in the understanding of small vessel disease, is that white matter intensities in the periventricular region, are probably due, not to arterial or arteriolar disease, but to venous disease. A close association has been observed between venous collagenosis and periventricular leukoaraiosis, both in Alzheimer disease and in elderly patients [25,26]. It is hypothesized that this venous wall disease may result in increased vascular resistance and hydrostatic vasogenic edema. Venous collagenosis was also observed in the CADASIL case mentioned above [20].

## 8. Opportunities for Future Research

Research is needed to assess differences among small vessels leading to small infarctions in the vascular centrencephalon, periventricular regions, and in lobar regions (sub-cortical regions over the convexity). Histological analyses with staining for amyloid, inflammatory markers, chemokines, and cytokines would be of interest. It would also be interesting to carry out imaging of amyloid and tau with positron emission tomography (PET), [27] and PET imaging for inflammation with fluorodeoxyglucose [28] in the region of white matter intensities seen on MRI in deep, periventricular, and lobar regions. Hachinski hypothesized (Invited lecture at the World Stroke Organization conference, October 2018) that low pressures in the lobar regions, as described by Blanco et al., may result in slower clearance of amyloid, resulting in aggravation of Alzheimer disease. 

## 9. Summary

Besides rare forms of small vessel disease such as vasculitits and CADASIL, there are probably three main forms of small vessel disease, that might be classified as deep/hypertensive, lobar/hypotensive, and periventricular/venous.

## 10. Conclusions

In considering the pathogenesis of cerebral small vessel disease, it is important to understand the very large pressure gradients in the brain. Understanding these differences should improve strategies for the prevention of cognitive decline by preventing stroke.

## Figures and Tables

**Figure 1 brainsci-09-00021-f001:**
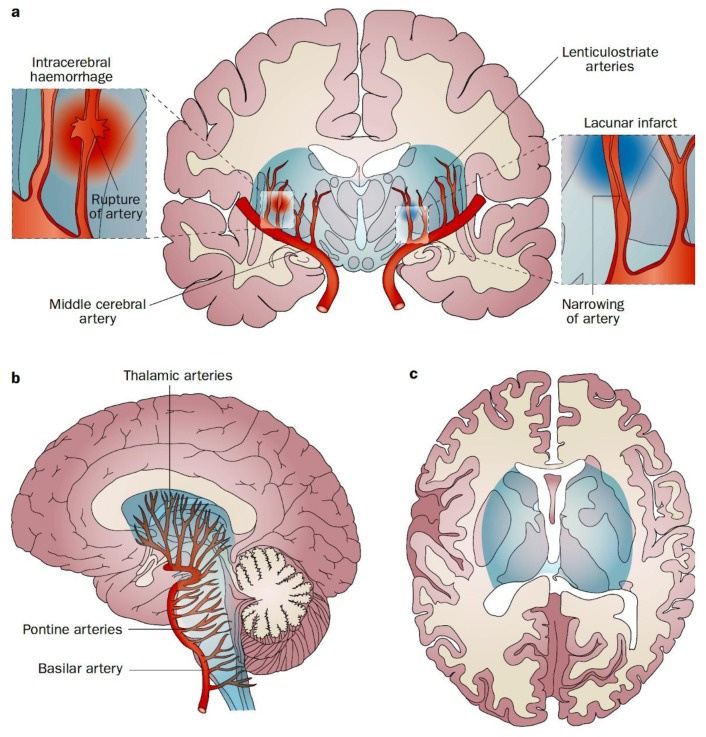
The Vascular Centrencephalon. Pathophysiology of lacunar infarction and haemorrhagic stroke. End arteries originate perpendicularly from the major vessels of the anterior and posterior circulation, without substantial collaterals from adjacent arteries. These arteries are particularly vulnerable to arterial hypertension, which causes fibrinoid degeneration and microaneurysms of the vessel wall, and results in narrowing (lacunar ischaemic infarct) or rupture (intracerebral haemorrhage) of arteries. End arteries supply the vascular centrencephalon (blue region), which represents phylogenetically older parts of the brain (including the brainstem, basal ganglia and thalamus), and adjacent white matter. (**a**) Coronal view. (**b**) Sagittal view. (**c**) Axial view. (Reproduced by permission of Nature from Sörös, P.; Whitehead, S.; Spence, J.D.; Hachinski, V. Antihypertensive treatment can prevent stroke and cognitive decline. *Nat. Rev. Neurol.*
**2013**, *9*, 174–178) [10].

**Figure 2 brainsci-09-00021-f002:**
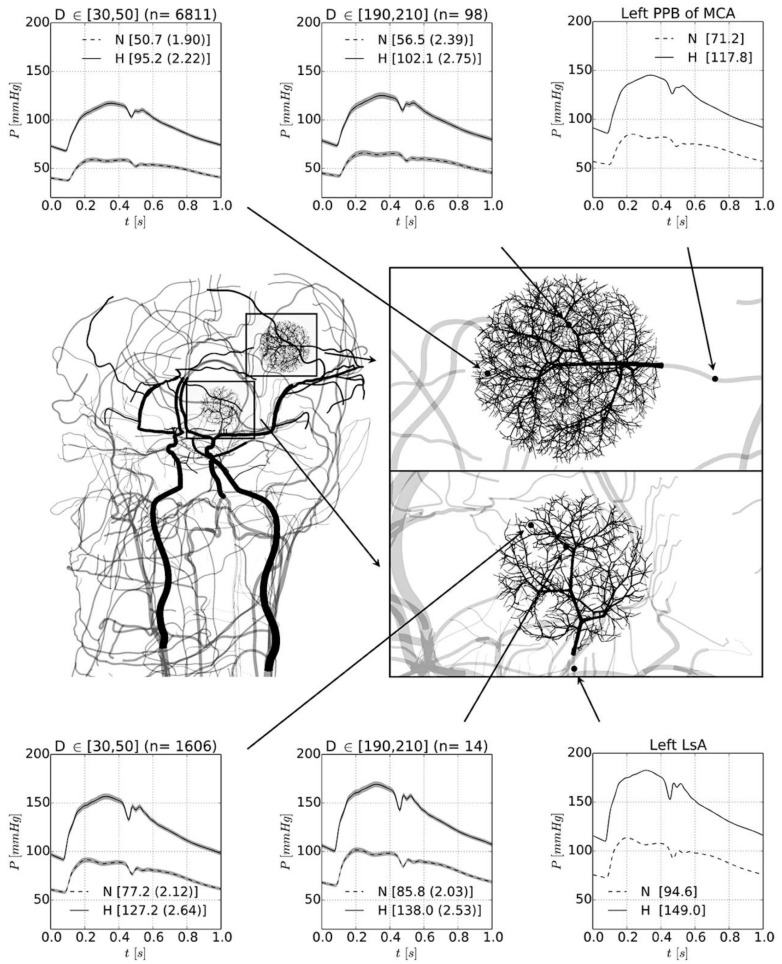
Blood pressure gradients in the brain. Detail of the peripheral beds corresponding to the lenticulostriate artery and to the posterior parietal branch of the middle cerebral artery. Pressure waveforms are shown for the normotensive (N, dashed line) and hypertensive (H, solid line) cases. Right panels (top and bottom) display the pressure waveform in the feeding artery to the corresponding arteriolar networks. Middle and left panels show the pressure level in arterioles with diameter ranges between D ∈ [190 μm, 210 μm] and D ∈ [30 μm, 50 μm], respectively; n indicates the number of vessels taken to calculate the average and SD pressure waveforms (grey-shaded area). In brackets, the mean arterial pressure is reported. LsA, lenticulostriate artery; MCA, middle cerebral artery; PPB, posterior parietal branch. (Reproduced by permission of BMJ from Blanco, P.; Mueller, L.; Spence, J.D. Blood pressure gradients in cerebral arteries: A clue to pathogenesis of cerebral small vessel disease. *Stroke Vasc. Neurol*. **2017**, *2*, 108–117) [11].

**Figure 3 brainsci-09-00021-f003:**
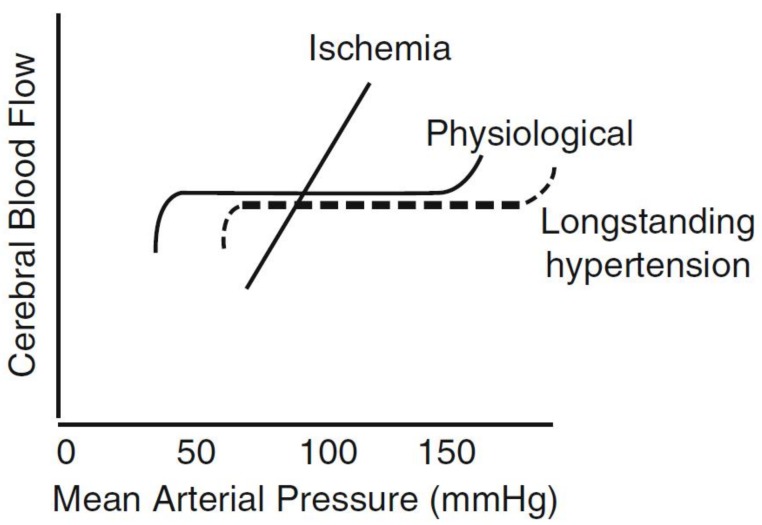
Auto-regulation of cerebral blood flow (CBF). In physiological conditions, CBF is auto-regulated over a wide range of perfusion pressures, from ~50 to 150 mmHg mean arterial pressure. This is shifted to the right in long-standing hypertension because of arteriolar hypertrophy. During acute ischemia, CBF becomes pressure passive, resulting in a marked reduction of CBF if the pressure drops too low. The threshold at which this becomes a problem will be higher for patients with long-standing hypertension, whose CBF auto-regulation is shifted to the right, because of arteriolar thickening. (Reproduced by permission Wolters Kluver from Spence, J.D. Treating hypertension in acute ischemic stroke. *Hypertension*
**2009**, *54*, 702–703) [24].

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
