# Peer review of "Blood Pressure Gradients in the Brain: Their Importance to Understanding Pathogenesis of Cerebral Small Vessel Disease"

_brainsci, 2019, doi:10.3390/brainsci9020021_

Reviewer 1 Report

Nicely written and succinct review of an under appreciated and recognised aspect of hypertensive small vessel disease. Well written with only few grammatical errors e.g. Pg 2, ln 51 - the rupture- these rupture.

I would like to see some more on future potential (modes of diagnostic approach, integrating biomarkers etc.) mentioned. No doubt, an increased understanding of  the pathogenesis of cerebral small vessel disease, it is important to understand the biorheological impact of the large pressure gradients in the brain, and the differentiation between the kinds of small vessel disease (deep/hypertensive, lobar/hypotensive, and venous/periventricular). Understanding these differences will improve strategies for prevention of cognitive decline by preventing stroke. It will also establish basic and translational studies to investigate the cellular and molecular underpinnings to lacunar infarctions.

Author Response

Nicely written and succinct review of an under appreciated and recognised aspect of hypertensive small vessel disease. Well written with only few grammatical errors e.g. Pg 2, ln 51 - the rupture- these rupture.

This has been corrected to ``they``.

I would like to see some more on future potential (modes of diagnostic approach, integrating biomarkers etc.) mentioned. No doubt, an increased understanding of  the pathogenesis of cerebral small vessel disease, it is important to understand the biorheological impact of the large pressure gradients in the brain, and the differentiation between the kinds of small vessel disease (deep/hypertensive, lobar/hypotensive, and venous/periventricular). Understanding these differences will improve strategies for prevention of cognitive decline by preventing stroke. It will also establish basic and translational studies to investigate the cellular and molecular underpinnings to lacunar infarctions.

A section has been added to suggest research opportunities arising from the understanding of blood pressure gradients in the brain.

Reviewer 2 Report

The author should include a statement of propose for the review – what knowledge gap is being filled here?

The key message appears to be that the entire cerebrovascular field has gotten off track with how it describes small strokes and that the term lacunar infarcts is being used inappropriately.  Focusing the review more on the idea that all cerebral arteries are not created equally would be beneficial because it could help correct the misinformation in the literature. 

There are several places where a simple definition would be helpful to the reader – for example, a definition of cerebral small vessel disease. Similarly, describing exactly what a cuff artifact is would be helpful to the reader.  Autoregulation should also be defined.

The review mentions only one animal study and the model used is not described particularly well.  I would recommend that the author take an extra sentence to describe the ET1 model before describing the effects of amyloid in the model.

The section on the misuse of the term lacunar infarction is a little repetitive.

The author comments that lower blood pressure recommendations are being contemplated – the AHA implemented the new lower BP recommendations more than a year ago. 

The description of the blood pressure gradients in the brain and their cause could be expanded. 

I would not recommend including the McEvoy paper about myocardial ischemia – it muddies the waters and will be confusing in a manuscript focused on the cerebral vasculature. 

The author states that patients with long standing hypertension have thickening of the cerebral arterioles – this needs to be clarified, I presume the author means that the arterioles have thicker walls.

Including a brief discussion of the gaps in the current research and the barriers that need to be overcome to move the field forward would be helpful to the reader and the scientific community. 

Author Response

Reviewer 2 comments

The author should include a statement of propose for the review – what knowledge gap is being filled here?

This has been added.

The key message appears to be that the entire cerebrovascular field has gotten off track with how it describes small strokes and that the term lacunar infarcts is being used inappropriately.  Focusing the review more on the idea that all cerebral arteries are not created equally would be beneficial because it could help correct the misinformation in the literature.

This has been added.

There are several places where a simple definition would be helpful to the reader – for example, a definition of cerebral small vessel disease.

Small vessel disease has been defined as ischemia due to disease of arterioles, as opposed to arterial disease or emboli.

 Similarly, describing exactly what a cuff artifact is would be helpful to the reader. 

Cuff artefact has been further described.

Autoregulation should also be defined.

A figure has been added to illustrate this.

The review mentions only one animal study and the model used is not described particularly well.  I would recommend that the author take an extra sentence to describe the ET1 model before describing the effects of amyloid in the model.

 The study has been described further.

The section on the misuse of the term lacunar infarction is a little repetitive.

The first reference to this has been deleted.

The author comments that lower blood pressure recommendations are being contemplated – the AHA implemented the new lower BP recommendations more than a year ago.

This has been clarified. The new guideline recommends treatment to below 130 mmHg, but in the wake of SPRINT a systolic target of 120 is being contemplated.

The description of the blood pressure gradients in the brain and their cause could be expanded.

 This has been expanded.

I would not recommend including the McEvoy paper about myocardial ischemia – it muddies the waters and will be confusing in a manuscript focused on the cerebral vasculature.

I disagree; it is another example of the same mechanism, in a different vascular bed, and clinically important.

The author states that patients with long standing hypertension have thickening of the cerebral arterioles – this needs to be clarified, I presume the author means that the arterioles have thicker walls.

This has been clarified.

Including a brief discussion of the gaps in the current research and the barriers that need to be overcome to move the field forward would be helpful to the reader and the scientific community.

A section has been added to suggest research opportunities arising from the understanding of blood pressure gradients in the brain.

Round  2

Reviewer 2 Report

There are just a few typographic errors that should be fixed (2 periods at the end of a sentence and capital letters where they are not needed).